# GENERATE EXPLORATIVE GOALS WITH LARGE LANGUAGE MODEL GUIDANCE

## ABSTRACT

Reinforcement learning (RL) struggles with sparse reward environments. Recent developments in intrinsic motivation have revealed the potential of language models to guide agents in exploring the environment. However, the mismatch between the granularity of environment transitions and natural language descriptions hinders effective exploration for current methods. To address this problem, we introduce a model-based RL method named Language-Guided Explorative Goal Generation (LanGoal), which combines large language model (LLM) guidance with intrinsic exploration reward by learning to propose meaningful goals. LanGoal learns a hierarchical policy together with a world model. The high-level policy learns to propose goals based on LLM guidance to explore the environment, and the low-level policy learns to achieve the goals. Extensive results on *Crafter* demonstrate the effectiveness of LanGoal compared to recent methods.

## 1 INTRODUCTION

Reinforcement learning has been widely used in decision-making tasks, but it struggles with long-horizon tasks and sparse reward settings. Especially in open-world tasks (Milani et al., 2020; Guss et al., 2021; Kanervisto et al., 2022), the agent needs to explore and make decisions to reach the goal in very large state space. Tasks like *obtain a diamond* in Minecraft, can involve long-horizon decision-making process and exploration for sparse reward signals, which significantly increase the difficulty of the task.

Given the intrinsic difficulty, reinforcement learning (RL) methods have been struggling to solve such tasks. Existing methods propose curiosity-driven exploration(Pathak et al., 2017; Ecoffet & Lehman, 2021), maximize disagreement between ensemble of models(Burda et al., 2019), or use intrinsic motivation(Schmidhuber, 1991; Pathak et al., 2017) to encourage the agent to explore the environment. Most of these methods give the agent a reward bonus when reaching unseen states, which can help the agent explore efficiently and avoid local optima. However, intrinsic reward methods can mislead the agent to favor meaningless noisy states or states with high transition uncertainty rather than reaching the goal, which leads to the inefficiency of the method in sparse reward settings.

Recently, with the rise of large language models (LLMs) and their ability as a few-shot learner (Achiam et al., 2023; Brown et al., 2020), they have been gradually used in decision-making tasks. Enriched with commonsense, LLMs can make reasoning and planning at abstract natural language level, break down the task into sub-tasks for downstream RL methods. LLMs can also provide promptable representation or exploration guidance with semantic meaning to the RL policy (Chen et al., 2024; Zhang & Lu, 2024), enabling the agent to make decisions with respect to the prompt. Thus, methods that combining LLMs with RL have been proposed to improve the performance of decision-making tasks.

However, the primary challenge lies in the combination of LLMs and RL methods, which requires a fast adaptation of the RL policy to the semantic meaning of environment state in an online manner. Existing works learn model-free policy with guidance from LLM, but lack of understanding of the semantic meaning. Thus, RL policy may not follow the guidance of LLMs or make a balance between reaching the LLM goals and exploration during the online training, which leads to the inefficiency of the method. Besides, RL policy may not be able to reach the goal proposed by LLMs when interacting with the environment, further compromising their effectiveness in goal-reaching tasks.

In this paper, we propose LanGoal, a model-based reinforcement learning method with hierarchical policy that combines with the LLM guidance efficiently. We claim that a hierarchical behavior is beneficial for the agent to solve this problem by setting a meaningful goal regarding the LLM guidance. Our method consists of a hierarchical policy training together with a world model. LLM gives semantic guidance to the high-level policy, which generates abstract actions as goals for the low-level policy as controller to reach. Inspired by recent advancement in controllable generation(Ho & Salimans, 2022; Dhariwal & Nichol, 2021) and its application in RL, we propose a novel method to combine the LLM guidance with the high-level policy to propose meaningful goals. This, as a result, improves the overall goal-reaching ability. We conduct extensive experiments to show the effectiveness of our method, compared with various baselines using different RL methods and LLMs. Our results reveal the potential of improving the performance on decision-making tasks combining LLMs and RL.

**Contributions.** The main contributions of this paper are as follows:

- We propose a novel model-based reinforcement learning method with hierarchical policy that combines with the LLM guidance efficiently.
- We introduce a new method to improve the effect of goal-reaching ability and inference performance at test time.
- We conduct extensive experiments on tasks in open-ended environment *Crafter* to show the effectiveness of our method, compared with various baselines using different RL methods and large language models.

## 2 RELATED WORKS

**Model-based RL.** Model-based RL(MBRL) methods learn a world model through online interactions or offline dataset (Ha & Schmidhuber, 2018; Hafner et al., 2020). Agent then learns a policy with the generated trajectories from interaction with the world model and improves the data efficiency. Existing works successfully apply MBRL methods in various domains including Atari games, locomotion tasks and open-ended environments, demonstrating the scalability of MBRL methods in decision-making tasks. (Hafner et al., 2023; 2021; 2019; Hansen et al., 2022; 2024). Lin et al. (2024) trains a multimodal world model using natural language descriptions and visual observations in the environment, enabling the agent to learn representations combining both modalities. We employ similar idea to learn multimodal embeddings for world model, while also consider incorporating the guidance from LLM using a hierarchical policy to improve exploration ability.

**Hierarchical reinforcement learning for exploration.** Hierarchical reinforcement learning offers a promising way to improve the exploration ability of RL methods, particularly in sparse reward settings. Hierarchical policy integrate effectively with intrinsic reward methods to facilitate temporal abstraction (Kulkarni et al., 2016; Gumbsch et al., 2023), design dense reward for agents to explore the environment (Steccanella et al., 2020; McClinton et al., 2021). Existing works also combines hierarchical policy learning with world model to improve the exploration ability of model-based RL methods. Hierarchical policy set random goals (Mendonca et al., 2021) or emply a divide-and-conquer-like strategy (Hamed et al., 2024) to explore the environment. Hafner et al. (2022) introduce a method to learn a hierarchical policy with intrinsic reward combines with world model, which helps the agent to explore in sparse reward settings. These methods typically utilize model uncertainty to encourage the agent to visit unseen states or transitions with high uncertainty.

However, such intrinsic rewards or heuristic methods can mislead the agent, such as favoring the states with high transition uncertainty rather than reaching the goal, which leads to the inefficiency of the method in sparse reward settings. Especially when meeting large state space and complex tasks, intrinsic reward methods may fail to guide the agent to reach the goal efficiently. In this work, we combine guidance from LLM with intrinsic reward, aiding the agent to explore the environment towards meaningful goals. We train a hierarchical policy to generate goals with aligned with LLM guidance, and try to explore and adhere to the guidance simultaneously.

**RL with LLM guidance.** Open-ended environments(Milani et al., 2020; Guss et al., 2021; Kanervisto et al., 2022; Hafner, 2021; Matthews et al., 2024) aresignificant due to their connections with reality. Tasks in open-ended environments, like *obtain a diamond*, can involve long-horizon decision making, which significantly increase the difficulty of the task. However, RL methods struggle with low sample

efficiency, especially when meeting sparse reward settings. Recent advancements in natural language processing with LLMs have garnered significant attention. LLMs such as GPT series(Brown et al., 2020; Achiam et al., 2023) are regarded as promising on decision making. LLMs are also highly expected to improve RL methods by offering semantic information and commonsense of the task (Chen et al., 2024; Zhang & Lu, 2024). One way is to give better representation or goals to the policy. P2RL(Chen et al., 2024) generates promptable representations for policy learning by visual question answering with environment observations. Zhang et al. (2023); Zhou et al. (2024) generate task image with the help of LLM as goal for the low level policy. Another way is reward shaping. LiFT(Nam et al., 2023) adjust MineClip reward by refining the description of current observation with MLLM. Zhang et al. (2024) compares different types including codes, preferences and goals on downstream RL methods. Prakash et al. (2023) train hierarchical policy as skills with LLM decide which skill to use next. While few of them have addressed the misalignment between the granularity of environment transitions and natural language descriptions, which can be less helpful when suitable language descriptions of transitions are unavailable in the environment.

## 3 PRELIMINARIES

We consider a partially observable Markov decision process (POMDP) defined by a tuple $(\mathcal{S}, \mathcal{A}, \mathcal{O}, P, R, \gamma)$, where $\mathcal{S}$ is the state space, $\mathcal{A}$ is the action space, $\mathcal{O}$ is the observation space, $P$ is the transition function, $R$ is the reward function, and $\gamma$ is the discount factor. The goal of the agent is to learn a policy $\pi$ that maximizes the expected return $\mathbb{E}_\pi[\sum_{t=0}^{\infty} \gamma^t r_t]$.

We further define a set of goals $\mathcal{G}$ that the agent can reach in the environment. These goals can be expressed in natural language or other forms of semantic information like embeddings, and we assume that for any two states $x_t, x_{t+h} \in \mathcal{O}$ with fixed interval $h$, the expression of the state changes can also be represented by natural language $f(x_t, x_{t+h}) = g_t^{\text{inv}} \in \mathcal{G}$. Given an observation $x_t \in \mathcal{O}$ and its language description $l_t$ at timestep $t$, LLM can decide a $g_t \in \mathcal{G}$ as goal for the policy to reach, then the RL policy $\pi$ takes $g_t$ and $o_t$ as input to make action $a_t$ in the environment until the next goal is proposed by LLM.

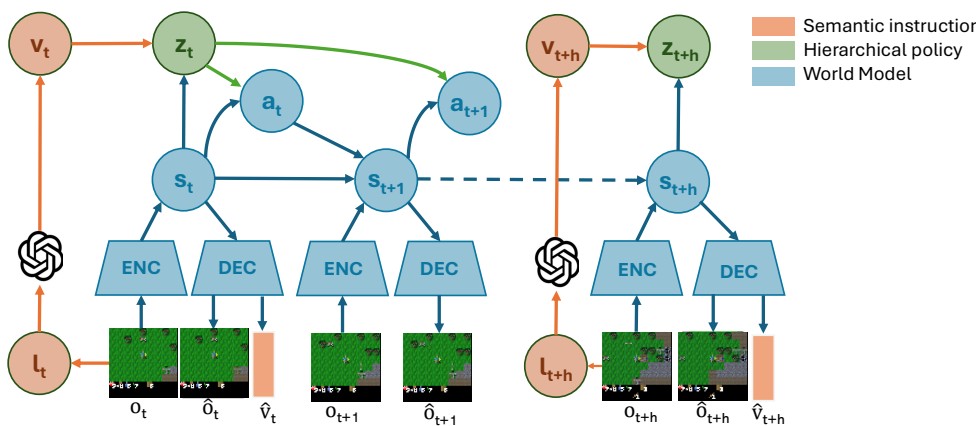

Figure 1: World model learning structure of LanGoal. Components like reward prediction are omitted for clarity. For every $H$ timesteps, the agent query LLM to obtain an embedded natural language goal $v_t$. The higher-level policy takes $s_t$ and $v_t$ as input to propose a goal $z_t$. The lower-level policy then generate a sequence of actions $a_t$ and interact with the environment until the next goal is proposed.

# 4 METHODS

In this section, we introduce the proposed method in detail. We first introduce how we prompt the LLM to generate skills for high-level policy, then we describe the world model and design of hierarchical policy. Finally, we introduce our method during test time to improve the goal-reaching ability.

## 4.1 PROMPTING LLM FOR GUIDANCE GENERATION

We query the LLM for a fixed timestep interval $H$ to ensure the responsed natural language goal is reachable for RL policy in the environment. Given the observation $o_t$ at timestep $t$, we first transform it into a natural language description $l_t$, which contains the necessary semantic information of the environment such as the inventory, location, and task description in the environment. Additionally, we employ a captioner to label the state changes in previous $H$ steps, showing which goal is actually reached by RL policy, denoted as $g_t^{\text{inv}}$ as its analogy to inverse dynamics. Then we prompt the LLM with $l_t$ to decide a goal $g_t$ to reach and use a pretrained encoder to transform $g_t$ into a vector $v_t$ for high-level policy as input. We also use the same encoder to transform $g_t^{\text{inv}}$ into a vector $v_t^{\text{inv}}$ as additional information to train the world model. The detailed design of the prompt, captioner and encoder are provided in Appendices B and C.

## 4.2 WORLD MODEL LEARNING

We basically follow previous works to use the Recurrent State-Space Model (RSSM) (Hafner et al., 2023) as the dynamics model and predict the next state, reward and terminal signal. However, we additionally predict the goal representation $v_t$ proposed by LLM and the goal representation that actually reached during the previous $H$ steps, denoted as $v_t^{\text{inv}}$. This can help leverage the information of LLM guidance, measure the semantic similarity between the proposed goal and the current state when training the policy using imaging with the world model. We refer to (Lin et al., 2024; Liu et al., 2024) to give a concise expression of the world model, consists several networks that are optimized jointly:

$$
\begin{aligned}
\text{Sequence model:} \quad & \hat{s}_t, h_t = \text{seq}_\theta \left( h_{t-1}, s_{t-1}, a_{t-1} \right) \\
\text{Encoder:} \quad & s_t \sim \text{enc}_\theta \left( s_t \mid h_t, o_t \right) \\
\text{Multimodal decoder:} \quad & \hat{x}_t, \hat{v}_t, \hat{r}_t, \hat{c}_t = \text{dec}_\theta \left( s_t, h_t \right) \\
& \hat{v}_t^{\text{inv}} = \text{dec}_\theta \left( s_{t-H}, s_t \right)
\end{aligned}
\tag{1}
$$

Where $h_t$ is recurrent state of the sequence model. The loss of the world model consists of the reconstruction loss, the prediction loss, and the reward loss. All loss terms are written as:

$$
\begin{aligned}
\text{Reconstruction Loss:} \quad & \mathcal{L}_x = \left\| \hat{x}_t - x_t \right\|_2^2, \\
& \mathcal{L}_v = \left\| \hat{v}_t - v_t \right\|_2^2, \\
& \mathcal{L}_v^{\text{inv}} = \left\| \hat{v}_t^{\text{inv}} - v_t^{\text{inv}} \right\|_2^2, \\
\text{Reward Loss:} \quad & \mathcal{L}_r = \text{catxent} \left( \hat{r}_t, \text{twohot} \left( r_t \right) \right), \\
\text{Continue Loss:} \quad & \mathcal{L}_c = \text{binxent} \left( \hat{c}_t, c_t \right), \\
\text{Prediction Loss:} \quad & \mathcal{L}_{\text{pred}} = \max \left( 1, \text{KL} \left[ \text{sg} \left( s_t \right) \| \hat{s}_t \right] \right), \\
\text{Regularizer:} \quad & \mathcal{L}_{\text{reg}} = \max \left( 1, \text{KL} \left[ s_t \| \text{sg} \left( \hat{s}_t \right) \right] \right),
\end{aligned}
\tag{2}
$$

where catxent is the categorical cross-entropy loss, binxent is the binary cross-entropy loss, sg is the stop gradient operator, KL refers to the Kullback-Leibler (KL) divergence. We then have total loss for the world model:

$$
\mathcal{L}_{\text{RSSM}} = \mathcal{L}_x + \mathcal{L}_v + \mathcal{L}_v^{\text{inv}} + \mathcal{L}_r + \mathcal{L}_c + \beta_1 \mathcal{L}_{\text{pred}} + \beta_2 \mathcal{L}_{\text{reg}},
\tag{3}
$$

in which $\beta_1 = 1.0$, $\beta_2 = 0.1$.

## 4.3 HIERARCHICAL POLICY

We design a hierarchical policy with two levels of policies to leverage LLM guidance for exploration. The low-level policy is a goal-reaching policy, try to reach the goal set by high-level policy. The high-level policy determines the goal state that meets both the LLM-proposed goal and the need to explore the environment. For simplicity, we synchronize the decision frequency of the high-level policy with that of the LLM, proposing a goal $z_t$ at every $H$ timesteps with high-level policy whenever the LLM proposes $g_t$. A different design of decision frequency is also feasible, which is left for future work.

**Goal autoencoder.** The goal state can be a high-dimensional continuous vector which is hard to make decisions for high-level policy. Thus, we use an autoencoder to transform the goal state into a discrete action space with lower dimension. The autoencoder compresses the state $s_t$ into high-level action space, and reconstruct the original state $\hat{s}_t$ from the given high-level action or compressed representation $u_t$. The reconstruct error is used to measure the novelty of the goal. Then we set $\text{dec}_\theta^H(z_t)$ as the goal for the low-level policy to reach. We refer to Hafner et al. (2022) to design the action space of high-level policy. Specifically, the goal encoder takes $s_t$ as input and predicts a matrix of 8×8 logits, samples a one-hot vector from each row, and flattens the results into a sparse vector with 8 out of 64 dimensions set to 1 and the others to 0. Gradients are backpropagated through the sampling by straight-through estimation (Bengio et al., 2013). The goal autoencoder is optimized end-to-end using the variational objective:

$$\mathcal{L}(\theta) = \left\| \text{dec}_\theta^H(z_t) - s_t \right\|^2 + \beta D_{\text{KL}}[\text{enc}_\theta^H(z_t \mid s_t) \| p(z)] \quad \text{where} \quad z_t \sim \text{enc}_\theta^H(z_t \mid s_t) \quad (4)$$

The components in hierarchical policy represent as:

$$
\begin{aligned}
\text{High-level Encoder:} \quad & u_t \sim \text{enc}_\theta^H(u_t \mid s_t) \\
\text{High-level Decoder:} \quad & \hat{s}_{t+h} \sim \text{dec}_\theta^H(\hat{s}_{t+h} \mid u_t) \\
\text{High-level policy:} \quad & z_t \sim \pi_\phi^H(z_t \mid s_t, v_t) \\
\text{Low-level policy:} \quad & a_t \sim \pi_\phi^L(a_t \mid s_t, \text{dec}_\theta^H(z_t))
\end{aligned}
\quad (5)
$$

**Reward design.** The high-level policy is encouraged to explore the environment towards the goal state generated by LLMs and try to reach a novel state in the meantime. When the high-level policy proposed a goal $z_t$, it receives an exploration reward with related to the reconstruction error between the future state $s_{t+H}$ and the decoded goal $\text{dec}_\theta^H(z_t)$, denoted as $r_{expl}$. The low-level policy is encouraged to reach the goal by maximizing the cosine similarity between the goal and current state as goal-reaching reward, denoted as $r_{goal}$. We also check if the goal proposed by LLM is reached or not and give guidance-following reward according to the cosine similarity of semantic guidance $v_t$ and $v_t^{inv}$, denoted as $r_{LLM}$. If the cosine similarity falls below 0.6, $r_{LLM}$ is set to 0 to ensure the policy's behavior correlates with $g_t$ and to prevent over-exploitation of this reward signal. Both rewards of high-level policy and low-level policy include the environment reward $r_t$ and the reward of reaching the goal by LLM $r_{LLM}$ to avoid misalignment between different levels of the policy. The reward items are written as:

$$
\begin{aligned}
r_{\text{expl}} &= \|\text{dec}_\theta^H(z_t) - s_{t+H}\|_2^2 \\
r_{\text{LLM}} &= \frac{v_t \cdot v_t^{\text{inv}}}{\|v_t\|\|v_t^{\text{inv}}\|} \quad \text{if} \quad \cos(v_t, v_t^{\text{inv}}) > 0.6 \quad \text{else} \quad 0 \\
r_{\text{goal}} &= \frac{\text{dec}_\theta^H(z_t) \cdot s_t}{\|\text{dec}_\theta^H(z_t)\|\|s_t\|} \\
r_{\text{high}} &= r_t + r_{\text{LLM}} + r_{\text{expl}} \\
r_{\text{low}} &= r_t + r_{\text{LLM}} + r_{\text{goal}}
\end{aligned}
\quad (6)
$$

**Actor-Critic Learning.** We use actor-critic learning to optimize the hierarchical policy and the critic and learn separate critic model for each component of the reward. We train the high-level policy and critic with abstract trajectories $\{\hat{s}_t, \hat{z}_t, \hat{s}_{t+H}, \hat{r}_{\text{high}}\}$ extracted from imagined trajectories generated by the world model. See details in Appendix D. Following the expression in (Lin et al., 2024; Liu et al., 2024), the actor and the critic give:

$$\text{Actor:} \quad \pi_\phi^H(z_t \mid s_t, v_t), \quad \pi_\phi^L(a_t \mid s_t, \text{dec}_\theta^H(z_t)) \quad \text{Critic:} \quad V_\psi^H(s_t), \quad V_\psi^L(z_t). \quad (7)$$

## 4.4 TEST-TIME TECHNIQUES

**Classifier-free guidance.** Classifier guidance (Dhariwal & Nichol, 2021) and classifier-free guidance (CFG) (Ho & Salimans, 2022) are first proposed for controllable generation. By conditioning the model on the classifier explicitly or implicitly, these methods approximate conditional score function in diffusion models and can generate samples that are more likely to be classified as the target class given a larger guidance scale. Without theoretical guarantee, CFG has also been used analogously for goal-reaching policy (Zhou et al., 2024) and generative models like conditional variational autoencoders (Lifshitz et al., 2024) and achieve better controllability. In this case, policy with CFG can be interpreted as a biased sampler, which favors goals that are more likely to be classified as the target class.

During test time, we also use the CFG policy $\pi_{CFG}$ on the higher-level policy to propose goals to check if the

---

**Algorithm 1** LanGoal

**while** acting **do**
  Step environment $o_t, r_t, c_t \leftarrow \text{env}(o_{t-1}, a_{t-1})$.
  *// Update goal with LLM and high-level policy*
  **if** $t \bmod H = 0$ **then**
    $g_t \sim \text{LLM}(g_t | o_l)$
    $v_t = \text{txt\_enc}(v_t \mid g_t)$
    $z_t \sim \pi_\phi^H(z_t \mid s_t, v_t)$.
  $a_t \sim \pi_\phi^L(a_t \mid s_t, z_t)$.
**while** training **do**
  Sample batch $\{x, a, r\}$ from replay buffer.
  Update world model
  Update high-level autoencoder
  *// Imaging*
  Imagine trajectory $\{\hat{s}_t, \hat{a}_t, \hat{r}_t, \hat{o}_t\}$ with world model.
  Predict rewards $\{\hat{r}_t, \hat{r}_{LLM}, \hat{r}_{goal}, \hat{r}_{expl}\}$ .
  Update high-level policy and critic with abstract trajectories.
  Update low-level policy and critic with imagined trajectories.
**while** testing **do**
  Step environment $o_t, r_t, c_t, g_t \leftarrow \text{env}(o_{t-1}, a_{t-1})$.
  *// Update goal if goal is reached or after H steps*
  **if** $t \bmod H = 0$ or $\cos(v_t, v_{t-H}^{\text{inv}}) > 0.9$ **then**
    $v_t = \text{txt\_enc}(v_t \mid \text{LLM}(g_t \mid o_l))$
    $z_t \sim \pi_{CFG}^H(z_t \mid s_t, v_t)$.
  $a_t \sim \pi_\phi^L(a_t | s_t, z_t)$.

---

high-level policy learns to propose goals following the LLM's guidance. CFG policy gives:

$$\pi_{CFG} = (1 + \lambda)\pi_\phi^H(z_t \mid s_t, v_t) - \lambda\pi_\phi^H(z_t \mid s_t, v_t = \emptyset) \tag{8}$$

Where $\lambda$ is a parameter to control guidance scale of condition, and $\emptyset$ represents the empty goal. Here we use the caption "no operation" as the empty goal, which means the agent is captioned as not reaching any goal between the interval of two LLM decisions. We set $\lambda = 4.0$ in our experiment.

**Adaptive goal-reset interval.** Lower-level policy may reach the goal set by LLM before the predetermined time interval while still trying to reach the continuous goal. To better utilize the LLM guidance when testing, we propose an adaptive goal-reset interval, allowing for the revision of goals established by the LLM during test execution.

Since we have trained the goal embedding predictor $v_t$ and $v_{t-H}^{\text{inv}}$ with the same timestep interval $H$, we can adjust the goal reset interval based on the cosine similarity between $v_t$ and $v_{t-H}^{\text{inv}}$. At each timestep, we calculate the cosine similarity between $v_t$ and $v_{t-H}^{\text{inv}}$ before the policy has taken action. If the similarity exceeds a preset threshold $\tau = 0.9$, we regard the goal has been reached during the past $H$ timesteps and subsequently reset the goal indicated by the LLM. We query LLM with current description of observation $l_t$ to obtain a new $v_t$ and set a new goal with $\pi_\phi^H$ for the lower-level policy to reach. Refer to Algorithm 1 for more details.

## 5 EXPERIMENTS

Our experiments mainly aims to evaluate the following aspects of our method: 1. our proposed method can improve the performance of decision-making tasks and make meaningful explorations. 2. our method can achieve better goal-reaching ability compared with the state-of-the-art methods.

## 5.1 Experimental Settings

**Environment.** The Crafter environment is a grid world that features pure pixel observation and discrete action space. Crafter is designed similarly as a 2D Minecraft, featuring a procedurally generated, partially observable world. The player's goal is to unlock the entire achievement tree by collecting items, crafting tools and defeating monsters. The player will obtain +1 reward for each achievement unlocked and +/- 0.1 reward for obtaining or losing health points.

Besides the trajectory reward, Crafter also consider the Crafter score as evaluation metrics, computed as $S \doteq \exp(\frac{1}{N} \sum_{i=1}^{N} \ln(1 + s_i)) - 1$, where $s_i \in [0; 100]$ is the agent's success rate of achievement $i$ and $N = 22$ is the number of achievements.

**Baselines.** We consider employing ELLM (Du et al. (2023)), Dynalang (Lin et al. (2024)) and AdaRefiner (Zhang & Lu (2024)) as baselines that include natural language information in RL methods. We refer to the results of Dynalang from (Liu et al., 2024). We also compare against:

- other baseline algorithms that do not utilize natural language in each environment from (Hafner, 2021), including PPO (Schulman et al. (2017)), Rainbow (Hessel et al. (2018)).
- recent method that only use LLM to make decisions, including SPRING (Wu et al. (2023)), Reflexion (Shinn et al. (2024)) and ReAct (Yao et al. (2023)) from (Zhang & Lu, 2024).

**LLM.** LanGoal use `gpt-4-turbo-2024-04-09` as LLM in our experiments. We cached outputs of LLM for each query regards to their necessary information and reuse them if meeting the same query again to help reduce the running time.

## 5.2 Results

We train our method on Crafter with 1M and 5M steps to match different settings of previous works. Table 1 shows the results comparing with baselines. Our method outperforms all the compared methods on score, indicating a greater success rate in accomplishing difficult tasks. Additionally, our test-time techniques further enhance the performance, achieving even higher scores. Figure 2 illustrates the success rate for each task trained with 1M steps, in comparison with DreamerV3. LanGoal excels on relatively hard tasks, e.g. "collect iron" and "make stone pickaxe". We also display the success rate of each task when trained after 5M steps in Figure 3, shown in appendix. Our method continues to maintain a higher success rate on these challenging tasks.

## 5.3 Ablation Study

We conduct ablation study to evaluate the effectiveness of each component of our method. The results are shown in table 2. We also record the proportion of reached goals from LLM in the last column for each setting.

**LLM Guidance.** To evaluate the effectiveness of LLM guidance, we compare the performance of our method with different size of LLMs. We use GPT-4(`gpt-4-turbo-2024-04-09`) and GPT-4o-mini(`gpt-4o-mini-2024-07-18`) to generate goals for the agent and evaluate the performance of our method, denoted as LanGoal and LanGoal(w/ 4o-mini) respectively in table 2. We observe slight performance drop after replacing LLM, but the results still surpass other RL methods. We also note that smaller LLM like GPT-4o-mini tends to generate more unreached goals regardless of current state, or simply choose goals to keep agent alive. While larger LLM like GPT-4 can make decision regarding to the current state and propose meaningful goals for the agent, indicating the importance of effective LLM guidance.

**Hierarchical Policy.** We compare the performance of our method with and without the hierarchical policy. In this setting, we still apply $r_{LLM}$ into the reward to encourage the agent to reach the goal proposed by LLM. The lower-level policy then takes the state $s_t$ and embeddings of natural language description $v_t$ as input, denoted as LanGoal(w/o Hier) in table 2. From the results, we observe that simply adding $r_{LLM}$ into the reward cause explicit performance drop on all metrics, validating the misalignment problem between the natural language description and the environment transition.

| Method | Score | Reward | Steps |
|---|---|---|---|
| **LanGoal** | **34.0**±0.3 | **14.1**±2.2 | 5M |
| AdaRefiner (w/ GPT-4) | 28.2±1.8 | 12.9±1.2 | 5M |
| AdaRefiner (w/ GPT-3.5) | 23.4±2.2 | 11.8±1.7 | 5M |
| ELLM | - | 6.0±0.4 | 5M |
| DreamerV3 | 32.9±0.5 | 13.7±2.5 | 5M |
| **LanGoal** | **23.8**±3.6 | **11.4**±2.4 | 1M |
| Achievement Distillation | 21.8±1.4 | 12.6±0.3 | 1M |
| Dynalang | 16.4±1.7 | 11.5±1.4 | 1M |
| AdaRefiner (w/ GPT-4) | 15.8±1.4 | 12.3±1.3 | 1M |
| PPO (ResNet) | 15.6±1.6 | 10.3±0.5 | 1M |
| DreamerV3 | 14.5±1.6 | 11.7±1.9 | 1M |
| PPO | 4.6±0.3 | 4.2±1.2 | 1M |
| Rainbow | 4.3±0.2 | 5.0±1.3 | 1M |
| SPRING (w/ GPT-4) | 27.3±1.2 | 12.3±0.7 | - |
| Reflexion (w/ GPT-4) | 11.7±1.4 | 9.1±0.8 | - |
| ReAct (w/ GPT-4) | 8.3±1.2 | 7.4±0.9 | - |
| Vanilla GPT-4 | 3.4±1.5 | 2.5±1.6 | - |
| Human Experts | 50.5±6.8 | 14.3±2.3 | - |
| Random | 1.6±0.0 | 2.1±1.3 | - |

Table 1: The results on Crafter. w/ test represents using test-time techniques in Section 4.4. We report mean and standard deviation of algorithm performance across 5 random seeds for LanGoal.

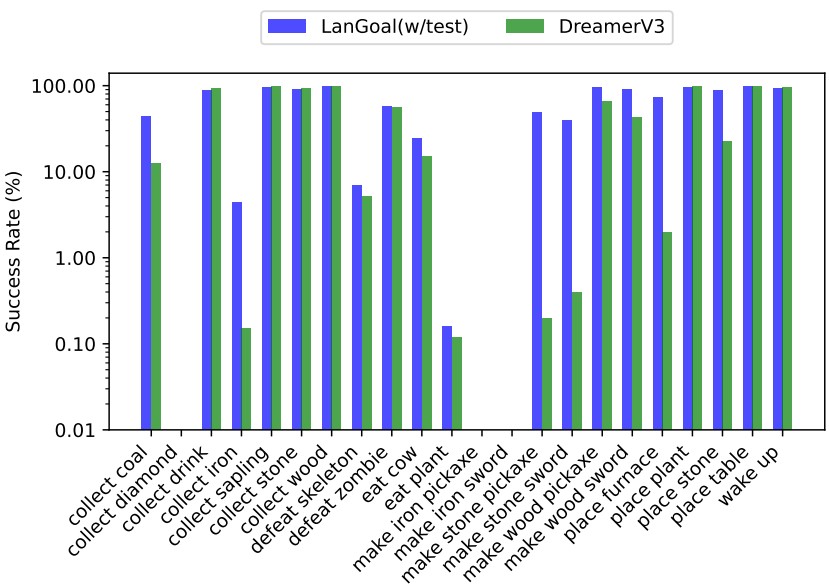

Figure 2: success rate on each task trained with 1M steps.

**Test-time Techniques.** We also compare the performance of our method with and without the test-time techniques. The results are shown in table 2. Besides the marginal performance gain, test-time techniques further improve the proportion of reached goals from LLM, shows that the high-level policy proposes goals following the LLM's guidance. As the high-level policy maximizes the goal-reaching reward and the guidance-following reward simultaneously and LLM may give unreachable guidance, some of the guidance may not be reached.

| Method | Score | Reward | Steps | Reached Goal |
|---|---|---|---|---|
| LanGoal(w/ test) | 34.3±1.0 | 14.1±2.3 | 5M | 40.5% |
| LanGoal | 34.0±0.3 | 14.1±2.2 | 5M | 38.7% |
| LanGoal(w/ test) | 24.5±3.8 | 11.6±2.3 | 1M | 44.0% |
| LanGoal | 23.8±3.6 | 11.4±2.4 | 1M | 43.7% |
| LanGoal(w/o Hier) | 19.6±2.9 | 10.7±1.3 | 1M | 41.7% |
| LanGoal(w/ 4o-mini) | 22.3±1.9 | 10.8±1.8 | 1M | 42.5% |
| DreamerV3 | 14.5±1.6 | 11.7±1.9 | 1M | - |

Table 2: Results of ablation studies. We report mean and standard deviation of each setting across 5 random seeds.

## 6 CONCLUSION

In this paper, we propose a novel method for decision-making tasks with language models, which is able to generate meaningful goals and reach them with high success rate. We also provide a novel test-time technique to improve overall performance of the model. Ablation studies on Crafter and demonstrate the effectiveness of each component of our method.

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

## A IMPLEMENTATION DETAILS

**Hyperparameters.** We keep most hyperparameters for world model learning and low-level policy learning the same as the (Hafner et al., 2023). For high-level policy, we test different sizes of action interval from {2,4,8} and find that 8 is a good trade-off between exploration and high-level policy training. When querying the LLM, we use its default hyperparameters. We test different CFG scale from {1.0, 2.0, 3.0, 4.0} and find that 4.0 provides the best performance.

| Hyperparameter | Value |
|---|---|
| Env steps | 5M |
| Imagination horizon $T$ | 15 |
| Train ratio | 512 |
| Batch size | 16 |
| Batch length | 64 |
| GRU recurrent units | 4096 |
| Decoder hidden units | 1024 |
| Decoder layers | 5 |
| $\text{enc}^H$ classes | 8 |
| $\text{enc}^H$ latents | 8 |
| $\pi^H$ action space | $8 \times 8$ |
| $\pi^H$ action interval | 8 |
| $\pi^H$ entropy $\eta$ | 0.5 |
| $\pi^L$ entropy $\eta$ | 3e-4 |
| LLM query interval | 8 |
| Similarity threshold | 0.6 |
| Goal-rest Similarity | 0.9 |
| CFG scale | 4.0 |

Table 3: Hyperparameters of LanGoal.

## B PROMPT DETAILS

We give the system prompt start by presenting the framework of the Crafter environment, employing Minecraft as an analogy. For each query, we extract necessary information from the observation and internal function of Crafter, including objects and creatures within the player's field of view, items in player's inventory, the player's health status and all goals should be reached. LLM then takes the system prompt and the information as input and output one goal for RL policy to reach. The following is a query example:

```
This is a game like minecraft.  Given the player's state, your
task is to choose the nearest goal the player can reach based
on your knowledge in minecraft.  The final purpose of player
is to keep player state healthy and finish all goals.  Answer
briefly with only one goal.  You can answer reached goal if
necessary.  Give your answer start with "goal".
Here is the player's state:

player state:  [player_state]
inventory:  [inventory]
reached goal:  [reached_goals]
unreached goal:  [unreached_goals]
nearby objects:  [objects]
```

## C CAPTIONER AND TEXT ENCODER

We categorized the transitions into the following types :

- subgoals. (e.g. collect iron, make stone pickaxe, wake up)
- other movements. (e.g. move up/down, no operation)

We use the internal information of Crafter environment to determine the type of the transition. When multiple subgoals is reached during the period, we caption the period as the less reached subgoal. We use SentenceBert all-MiniLM-L6-v2 (Wang et al., 2020) as the text encoder.

## D    ACTOR-CRITIC LEARNING

The actor aims to maximize the cumulative returns, i.e.,

$$R_t \doteq \sum_{\tau=0}^{\infty} \gamma^\tau (r_{t+\tau}). \tag{9}$$

Here $r_{t+\tau}$ represents the respective rewards of the low-level policy and the high-level policy at time step $t + \tau$. Then the bootstrapped $\lambda$-returns Sutton & Barto (2018) could be written as:

$$R_t^\lambda \doteq r_t + \gamma c_t \left( (1-\lambda)V_\psi(s_{t+1}) + \lambda R_{t+1}^\lambda \right), \qquad R_T^\lambda \doteq V_\psi(s_T). \tag{10}$$

The actor and the critic are updated via the following losses:

$$\mathcal{L}_V = \text{catxent}\left(V_\psi(s_t), \text{sg}\left(\text{twohot}\left(R_t\right)\right)\right),$$
$$\mathcal{L}_\pi = -\frac{\text{sg}\left(R_t - V(s_t)\right)}{\max(1, S)} \log \pi_\phi\left(a_t \mid s_t\right) - \eta \text{H}\left[\pi_\phi\left(a_t \mid s_t\right)\right]. \tag{11}$$

where $S$ is the exponential moving average between the 5th and 95th percentile of $R_t$, $H$ is the entropy of the policy.

## E    MORE RESULTS

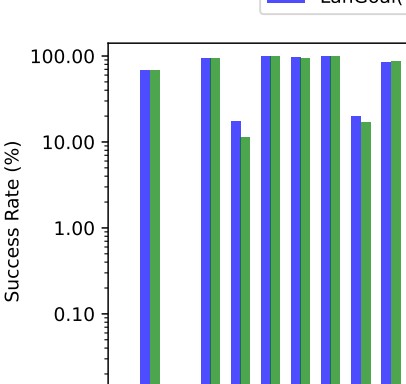

Figure 3: success rate for each task trained with 5M steps. LanGoal still performs well on hard tasks like *make iron pickaxe* and *make iron sword*.

