# OpenReview forum: "Generate explorative goals with large language model guidance"
_ICLR.cc/2025/Conference — ICLR 2025 Conference Withdrawn Submission_

### Official Review · Reviewer_4zna · 2024-10-22

**Soundness:** 2
**Presentation:** 1
**Contribution:** 3
**Rating:** 3
**Confidence:** 4

**Summary:**

This paper presents a new combination of goal-directed exploration supported by LLMs and Dreamer-like model-based deep RL. It shows impressive performance improvement in Crafter, a hard exploration RL task.

In its current state, the paper is poorly written and hard to parse. The text should be passed into a grammar checker, there are a lot of language errors throughout, which often makes comprehension difficult (some errors listed below but many more in the text). I suggest you use language tools and/or ask for the help of a native speaker colleague because I suspect improving the language would make the contribution a lot clearer.

I also find the approach to not be clearly motivated and not clearly situated in the space of the closest existing methods (Dreamer, ELLM, OMNI), see comments below. The novelty of the contribution could be made more explicit, because in the current state the approach seems to be a rather simple combination of Dreamer and goal-directed exploration with LLMs like Omni or ELLM.

The methods could also be more clearly explained with better descriptions of each of the variables involved in the many equations and losses presented here. Each should also be intuitively motivated. I'm not sure what the goal encoder is doing, how the high-level policy is trained here and how the LLM suggested goals are incorporated into the architecture.

I strongly encourage the authors to make considerable efforts in improving on the points listed above over the rebuttal period. I am inclined to augment my rating based on improvements on writing, motivations / positioning and methods descriptions.

**Strengths:**

The results seem impressive, Crafter is a hard task.

**Weaknesses:**

My questions and comments are listed below, but generally:
* the paper is poorly written with many grammatical errors which make comprehension hard
* methods not always clearly described, in particular about the high-level policy, the goal encoder and how the LLM-generated goals influence the architecture
* unsupported claim that the test-time technique help performance -- they do not seem to help
* only tested in one environment. If this paper is an extension of Dreamer then it would be good to test on a similar range of environments

**Questions:**

Related work:
* it's not so clear in the text why HRL or world models would help exploration, could this be explained in more details?
* Goal-based exploration is a whole field, not limited to its use in hierarchical RL. It's also a kind of intrinsic motivation (called competence-based in Oudeyer and Kaplan's framework (https://pubmed.ncbi.nlm.nih.gov/18958277/)), see a recent review of these approaches in https://arxiv.org/pdf/2012.09830
* In addition to ELLM there is another work using LLMs to generate exploration goals for RL in Crafter: OMNI (https://arxiv.org/pdf/2306.01711).  As far as I understand, the contribution of this paper is to combine this approach with a Dreamer-like model-based RL.
* It would be good to have a paragraph clearly listing similarities and differences between the proposed approach and the closest existing ones like Dreamer, Omni and ELLM. What are the real contributions and novelty here?
* Intrinsic motivation literature:
  * the list in the second paragraph is very sparse, all these methods are "intrinsic motivation," "curiosity-driven" could also apply to all methods cited here, and I think should be defined as "intrinsic motivation driving to explore" in opposition with intrinsic motivations driving to homeostasis (eg minimizing surprise).
  * Not all intrinsic reward methods are susceptible to the noisy TV issue.

Motivations:
* I'm not sure I understand the 4th paragraph of the intro. What is meant by "Existing works learn model-free policy with guidance from LLM, but lack of understanding of the semantic meaning" or "make a balance between reaching the LLM goals and exploration during the online training, which leads to the inefficiency of the method"?
* what is a "semantic guidance"? By the end of the intro, this term has still not been defined. Maybe an example could help the reader understand what is meant by this term?
* The second contribution also sounds vague, I'm not sure I understand what it refers to.
* Overall I think the text doesn't motivate the approach very well because it does not clearly link the contributions of this paper (not very clear in the intro) to the challenges posed by hard exploration problems and doesn't explain how the proposed approach aims to tackle them. This discussion should be made in light of existing approaches using LLMs for exploration like OMNI (see above) and ELLM. The first is not discussed at all while the second is only mentioned in the baseline but does not appear neither in the related work nor in the introduction.

Methods:
* h_t is not defined anywhere and does not appear in the graph, I assume it's the memory state from the RSSM?
* o_t appears as x_t in the Figure
* One of the appeals of model-based RL is that it's goal agnostic and thus can be reused for new goals. But here it seems like the world model is made goal-dependent by adding the auxiliary loss of predicting the agent's goal v_t from the internal states? Could you say more about why it's important to do that?
the multi-modal decoder seems to be split in two networks with different inputs but that have the same weights theta, what is going on here? * Maybe these are different networks with different weights?
* Algorithm 1: to define a new variable it's either x ~ distrib(x | y) or x = f(y), but here the authors use a combination of the two that seems odd: eg, v_t = txt_enc(v_t | LLM(g_t |o_l)), here txt_enc is just a function and the LLM can be sampled so maybe g_t ~LLM(g_t | o_l)  and then v_t = txt_end(g_t)?
* I'm not sure I understand what's going on with the Classifier-free guidance, but I also think it should be removed from the paper since it doesn't bring any significant performance improvement here.


Results
* Contrary to what is claimed in section 5.2, the addition of the test-time technique doesn't seem to add any significant boost (biggest difference is .7 with an std of 3.6). If the method doesn't help, what is a good reason to keep it in the paper? Should it be moved to the appendix?

Goal encoder:
* I do not understand what is being implemented here.
* "we use an autoencoder to transform the goal state into a discrete action space with lower dimension." → up to now the goal was a semantic embedding, and I don't understand why we would compress a goal into an action?
* the following sentence refers to the inner state representation of the RSSM (s_t), which is not the goal?
* The equation refers to z_t as the latent code, and s_t as the input, so I'm confused about what the "goal" is here.
* In the following equation 5, now z_t is replaced by u_t?
* I'm not sure what is going on here and this seems to be a key point in the algorithm.

---

> ### Author Response · Authors · 2024-11-24
>
> We thank the reviewer for the thoughtful feedback on our paper. We appreciate your detailed analysis and constructive suggestions.
>
> > motivation and existing methods.
>
> We thank the reviewer for the providing the related papers. We will clarify the motivations, related works and comparisons with existing approaches in future version.
>
> > Method details.
>
> We have fixed the unclear notations in method details and algorithm 1. We will elaborate the design and motivation of the method in a future version.
>
>
> > Results.
>
> We agree that the test-time technique only brings marginal performance gain. We remove it from Section 5.2, and keep it in Section 5.3 for ablation study, to show whether the high-level policy can generate goal states regards to the LLM instructions.

---

### Official Review · Reviewer_kseo · 2024-10-28

**Soundness:** 2
**Presentation:** 1
**Contribution:** 2
**Rating:** 3
**Confidence:** 4

**Summary:**

This paper introduces LanGoal, a model-based reinforcement learning (RL) framework that combines large language model (LLM) guidance with intrinsic exploration rewards to address the challenge of sparse rewards in reinforcement learning environments. LanGoal's hierarchical policy structure allows the high-level policy, informed by LLM guidance, to generate meaningful exploration goals, while the low-level policy aims to achieve these goals. The paper presents extensive experimental results on the Crafter environment, showing that LanGoal outperforms other methods in goal-reaching and exploratory efficiency, indicating its potential to enhance RL agents' performance in complex, sparse-reward tasks.

**Strengths:**

It is an interesting paper which combines large language model (LLM) guidance with model-based reinforcement learning (RL) to tackle sparse-reward environments. The strengths of this paper include:
   - This approach provides a novel perspective by using LLMs to guide exploration goals, addressing sparse reward in RL.
   - LanGoal’s hierarchical policy structure allows for LLM-guided high-level goal generation and low-level goal achievement, offering a well-structured and adaptable approach that brings interpretability to RL through language-based goals.
- The hierarchical approach, where the high-level policy sets goals based on LLM guidance and the low-level policy works to achieve them, is well-articulated and theoretically sound. The paper’s clear structure and mathematical formulations make it easy to follow and reproduce.
   - The authors provide thorough experiments in the Crafter environment, comparing LanGoal with state-of-the-art methods in terms of success rate and goal achievement along with ablation studies.
   - By focusing on sparse-reward settings, LanGoal has implications for applications requiring efficient exploration in large state spaces, such as robotics, autonomous navigation, and open-world gaming. The use of language guidance could provide intuitive and adaptable control strategies for these domains.

**Weaknesses:**

While the paper introduces a promising framework with LanGoal, there are still some major weaknesses which need to be resolved:

-  **Limited Novelty:** One major weaknesses is the paper lacks of novelty. While the paper provides a well-executed framework, its novelty may be limited due to reliance on existing concepts within reinforcement learning and large language model (LLM) integration. Specifically:
   - The hierarchical structure that combines high-level goal-setting with low-level goal-reaching policies is a common technique in reinforcement learning, and several works have explored using language guidance for RL exploration. LanGoal’s approach, while effective, primarily extends these existing ideas without introducing fundamentally new mechanisms or problem formulations.
   - The combination of model-based RL with LLM-guided exploration goals appears to be an incremental improvement, building upon previously proposed methods rather than proposing a substantially different or unique approach.

- **Limited Environment Generalizability:** The experiments are conducted solely on the Crafter environment, which, while relevant, may not fully represent the diversity and complexity of sparse-reward or open-world environments. The reliance on a single environment limits the ability to generalize the findings and assess the robustness of LanGoal. This is one of the major weaknesses of this paper. Including additional environments, such as navigation-based tasks or open-world settings (e.g., Minecraft or other 3D environments), could provide stronger evidence for LanGoal’s adaptability and effectiveness.

- **Dependency on Predefined Language Goals:** The method depends heavily on predefined language-based goals generated by an LLM, which may not fully align with the agent’s state or context in more dynamic environments. If the LLM fails to propose contextually appropriate goals, it could reduce the model's efficiency and reliability. Incorporating a feedback mechanism where the agent’s exploration results influence subsequent goal generation could make the approach more adaptive. Alternatively, investigating the LLM’s goal adaptability across different environments could clarify its limitations.

- **Writing and Layout Issues:** The paper’s readability is compromised by several writing and layout issues, affecting its clarity and overall presentation. Frequent instances of excessive blank spaces between figures and equations disrupt the reading flow, making the document appear less polished and professional.

- **Clarity on Hierarchical Policy Learning:** The hierarchical policy’s training process, particularly the interaction between high-level and low-level policies, could be clarified further. It is not fully clear how the policies handle conflicting objectives between goal exploration and goal-reaching under varying reward structures. Providing a more detailed breakdown of the policy training dynamics, possibly with ablation studies on the hierarchical structure itself, could enhance understanding and reproducibility.


- **Limited Discussion on Failures or Negative Results:** The paper predominantly presents positive findings but lacks an in-depth analysis of failure cases or scenarios where LanGoal may struggle, such as instances where LLM guidance contradicts optimal exploration paths. Adding a section on observed limitations or failure modes, and discussing potential mitigations, would improve transparency and provide a balanced perspective on the framework’s performance.

- **Scalability Concerns with LLM Use:** The use of an LLM for continual goal guidance raises potential scalability and efficiency concerns, particularly in larger or real-time environments where frequent LLM queries could lead to computational bottlenecks. Exploring lightweight alternatives, such as using distilled or smaller models that capture core semantic information, or reducing the query frequency with effective goal-persistence strategies, may mitigate scalability issues.

**Summary of Weaknesses**

While the paper introduces a promising framework with LanGoal, it faces significant challenges that limit its readiness for publication. The most critical issue is limited novelty; the approach largely builds on established techniques in reinforcement learning (RL) and large language model (LLM) guidance without offering new problem definitions or mechanisms. Additionally, limited environment generalizability is a concern, as the experiments are conducted solely on the Crafter environment, restricting the evidence for LanGoal's broader applicability. Along with other weaknesses, it collectively suggests that the paper requires substantial revisions. It is not ready for publication.

**Questions:**

See above sections.

---

> ### Author Response · Authors · 2024-11-24
>
> We thank the reviewer for the thoughtful feedback on our paper. We appreciate your detailed analysis and constructive suggestions.
>
> > Limited Novelty.
>
> It is true that our method is based on existing concepts. The main novelty of our paper is to combine the hierarchical policy with the language-based goal generation, and train this policy with world model. In this way, we reduce the query frequency of LLM, and enable goal state generation follows LLM instructions.
>
> > Limited Environment Generalizability.
>
> We will test our method on more environments on other navigation tasks and open-world tasks like Minecraft in the future work.
>
> > Dependency on Predefined Language Goals.
>
> Our method uses predefined, limited language instructions for high-level policy to produce goal states after considering the cost of LLM queries and the setting of the Crafter environment.
>
> It would be interesting to discover complex and meaningful behaviors other than predefined instructions using LLM, but this would require a much larger amount of LLM queries and is beyond the scope of this paper.
>
>
> > Limited Discussion on Failures like LLM guidance contradicts optimal exploration paths.
>
> We thank the reviewer for the constructive suggestion.
> At current stage, effects of failure cases can only shown by using smaller LLMs(e.g. gpt-4o-mini) in Sec. 5.3, which provide trivial or unreachable instructions, lower the performance.
> We will explore the effect of inefficient LLM instructions on exploration in future work.
>
> > Scalability Concerns with LLM Use.
>
> We agree that smaller models is crucial on saving the cost of RL training with LLM. However, as far as we have tested, most smaller LLM includes gpt-4o-mini fails to make complex decisions based on the current state, which restricts the application of smaller LLMs. Although we have make an attempt to reduce query frequency by using hierarchical policy, it is still an open problem to further reduce the cost.

---

> ### Comment · Reviewer_kseo · 2024-11-25
>
> Thanks for authors' response. I'll keep my score.

---

### Official Review · Reviewer_UyTj · 2024-10-31

**Soundness:** 1
**Presentation:** 1
**Contribution:** 1
**Rating:** 1
**Confidence:** 5

**Summary:**

This paper proposes LanGoal.

**Strengths:**

None

**Weaknesses:**

This article bears a strong resemblance to an article on arXiv: "World Models with Hints of Large Language Models for Goal Achievement" (DLLM), https://arxiv.org/abs/2406.07381. I suspect there is plagiarism involved.

The most evident similarity lies in the methodology and the corresponding equations. In Section 4.2 of "LanGoal," all the equations are identical or nearly identical to those in Section 4.3 of DLLM, including specific abbreviations such as "catxent" (categorical cross-entropy loss) and "binxent" (binary cross-entropy loss), which are not conventional in machine learning. In the "reward design" section of 4.3 in "LanGoal," the only apparent modification from Section 4.2 of DLLM is changing the threshold hyperparameter M from 0.5 to 0.6.

"LanGoal" claims to provide further details about the high-level and low-level critics in Appendix D, but the appendix only replicates the latter part of Section 4.3 from DLLM paper, without any changes or additional information. In Section 4.3 and Figure 1, "LanGoal" introduces a high-level policy that generates a hidden state and a low-level policy that produces the actual action. However, this is essentially the same as the encoder in the world model of DreamerV3, the backbone algorithm of DLLM, with the only difference being that "LanGoal" treats the hidden state as an action, while it is viewed as a latent variable that captures information from both the history and current state.

In Section 4.4 of "LanGoal," the authors cite two papers on diffusion models to introduce the concept of guidance, claiming to have implemented a related classifier-free guidance (CFG) policy. However, they do not provide any details about the method or its implementation, such as the models used or relevant hyperparameters. Since their approach essentially replicates DLLM,  this method does not require diffusion models, as RSSM and diffusion are fundamentally different frameworks. This seems like an awkward attempt to artificially broaden the scope of their work.

Moreover, the structure of "LanGoal" bears a strong resemblance to DLLM paper, including a similar depiction of the algorithm (Figure 1), a nearly identical format for presenting prompt details in the appendix, and comparable content and formatting in the experimental section, particularly in Table 1 (Section 5.2).

Besides, all baselines experiment results are the same as DLLM paper without citing that paper.

**Questions:**

None

---

> ### Author Response · Authors · 2024-11-24
>
> We thank the reviewer for their feedback on our paper.
>
>
> > 4.2 of "LanGoal," all the equations are identical or nearly identical to those in Section 4.3 of DLLM, including specific abbreviations such as "catxent" .
>
> We apologize for missing the reference since we only predict additional objects compared with RSSM of DreamerV3 [1].
>
> We find that **DLLM use exactly same equations with Dynalang [2] without reference, so we add both references** in Section 4.2 in our revised version.
>
>
> > Reward design.
>
> We use similar reward design(thresholded cosine similarity) as DLLM only on the part of language similarity. However, the entire reward function is different from DLLM. In specific, we add the reward together with the goal-reaching reward and the exploration reward, give the reward to the high-level policy for every $h$ steps, while DLLM multiplies the language similarity reward with the exploration reward for different LLM instructions every step.
>
> In this way, we do not hinder the exploration of the high-level policy when LLM instructions difficult to achieve, while the policy can obtain higher rewards when following the LLM instructions, encouraging the high-level policy to learn to generate goal states regards to the LLM instructions.
>
> > Actor and critic.
>
> We apologize for missing the reference since we do not make changes on the actor and critic learning of DreamerV3 [1].
>
> We find that **DLLM use basically same equations with Dynalang [2] without any reference or explanation, so we add both references** in our revised version in Appendix D.
>
> > Encoder-decoder for high-level policy.
>
> We suggest the reviewer to refer to Section 4.3 of our paper. The encoder-decoder is used to generate goal states regards to the LLM instructions, which can hardly be similar from design to the encoder of DreamerV3.
>
>
> > CFG policy.
>
> We suggest the reviewer to refer to Section 4.4 of our paper for the implementation and relevant hyperparameters of the CFG policy.
>
> The CFG policy in Section 4.4 is not related with RSSM, but works only on the high-level policy.
> Since CFG is actually a biased sampling process when guidance strength $\lambda \neq 1$ for diffusion models and non-diffusion models, we only use it as a sampling method to show that the high-level policy actually learns to generate meaningful goals regards to the LLM instructions.
>
> In detail, we use guidance scale $\lambda = 4$ and use 'no achievement is reached' as the unconditional case.
>
> > Algorithm figure and prompt details in the appendix.
>
> There is few evidence to support this claim. As we mentioned, the algorithm basically hopes generate goal states regards to the LLM instructions with LLM instructions, which is refleted in the figure.
>
> The prompt details in both work generally require the LLM to 'choose subgoal given the key infomation'. While DLLM requires LLM to choose 5 subgoals , while ours requires LLM to choose only 1 subgoal, so that the high-level policy can learn to generate goal states corrsponding to the LLM instructions.
>
> As supplementary details, Given the view of the agent, we choose all items in 3\*3 grid with agent's position as the center to generate the 'nearby objects' in prompt details, ignoring the landform like 'water' and 'path'.
>
> > Baseline experiments and Table 1(Section 5.2).
>
> We apologize for missing the reference of baseline experiments.
> We have added the reference in the revised version, with results of LLM based methods and AdaRefiner from AdaRefiner, RL methods from Crafter benchmark, PPO (ResNet) and Dynalang from DLLM [3].
>
> We would like to clarify that, our work and DLLM differ in motivation and respective method on processing the LLM instructions. Our work and DLLM both consider exploration in sparse reward environments, but our work also aims to generate goal states regards to the LLM instructions. We thus query LLM to only generate 1 instruction each time, and add the exploration reward with LLM reward to encourage the high-level policy to generate goal states regards to the LLM instructions while exploring the environment.
>
> References:
>
> [1] Danijar Hafner et al. "Mastering diverse domains through world models." arXiv preprint arXiv:2301.04104, 2023.
>
> [2] Jessy Lin et al. "Learning to model the world with language." In Forty-first International Conference on Machine Learning, 2024.
>
> [3] Liu, Zeyuan, et al. "World Models with Hints of Large Language Models for Goal Achieving." arXiv preprint arXiv:2406.07381 (2024).

---

> ### Comment · Area_Chair_9MmJ · 2024-11-25
> **Please read rebuttal**
>
> Dear Reviewer UyTj, Could you please read the authors' rebuttal and give them feedback at your earliest convenience? Thanks. AC

---

### Official Review · Reviewer_hsev · 2024-11-02

**Soundness:** 1
**Presentation:** 1
**Contribution:** 1
**Rating:** 1
**Confidence:** 5

**Summary:**

This paper introduces a model-based RL method named Language-Guided Explorative Goal Generation (LanGoal) that combines large language model (LLM) guidance with intrinsic exploration reward to propose meaningful goals for exploration in sparse reward environments. LanGoal consists of a hierarchical policy that generates goals based on LLM guidance and a world model that learns the dynamics of the environment. The paper provides extensive experimental results on the Crafter environment to demonstrate the effectiveness of LanGoal compared to other methods.

**This paper shows significant similarities to the arXiv article "World Models with Hints of Large Language Models for Goal Achievement" (DLLM), available at https://arxiv.org/abs/2406.07381, raising concerns about potential plagiarism.**

**Strengths:**

none

**Weaknesses:**

**This paper shows significant similarities to the arXiv article "World Models with Hints of Large Language Models for Goal Achievement" (DLLM), available at https://arxiv.org/abs/2406.07381, raising concerns about potential plagiarism.**
- The equations in Section 4.2 of "LanGoal" are identical or nearly identical to those in Section 4.3 of the DLLM paper, including specific terms like "catxent" and "binxent."
- The "reward design" section of "LanGoal" only differs from DLLM by a slight change in the threshold hyperparameter from 0.5 to 0.6.
- Appendix D of "LanGoal" merely replicates part of Section 4.3 from DLLM without any additional information or changes.
- The description of a high-level policy generating a hidden state and a low-level policy producing actions in "LanGoal" closely mirrors the encoder concept in DreamerV3, the backbone of DLLM, with minimal differences.
- In Section 4.4, "LanGoal" mentions implementing a classifier-free guidance (CFG) policy but fails to provide specific details about the implementation or relevant hyperparameters.
- The results for baseline experiments in "LanGoal" match those in DLLM without proper citation.

**Questions:**

none

---

> ### Author Response · Authors · 2024-11-24
>
> We thank the reviewer for their feedback on our paper.
>
> > 4.2 of "LanGoal," all the equations are identical or nearly identical to those in Section 4.3 of DLLM, including specific abbreviations such as "catxent".
>
> We apologize for missing the reference since we only predict additional objects compared with RSSM of DreamerV3 [1].
>
> We find that **DLLM use exactly same equations with Dynalang [2] without reference, so we add both references** in Section 4.2 in our revised version.
>
> > Reward design.
>
> We use similar reward design(i.e., thresholded cosine similarity) as DLLM only on the part of language similarity. However, the entire reward function is different from DLLM. In specific, we add the reward together with the goal-reaching reward and the exploration reward, give the reward to the high-level policy for every $h$ steps, while DLLM multiplies the language similarity reward with the exploration reward for different LLM instructions every step.
>
> In this way, we do not hinder the exploration of the high-level policy when LLM instructions difficult to achieve, while the policy can obtain higher rewards when following the LLM instructions, encouraging the high-level policy to learn to generate goal states regards to the LLM instructions.
>
> > Appendix D of "LanGoal" merely replicates part of Section 4.3 from DLLM without any additional information or changes.
>
> We apologize for missing the reference since we do not make changes on the actor and critic learning of DreamerV3 [1].
>
> We find that **DLLM use basically same equations with Dynalang [2] without any reference or explanation, so we add both references** in our revised version in Appendix D.
>
> > The description of a high-level policy generating a hidden state and a low-level policy producing actions in "LanGoal" closely mirrors the encoder concept in DreamerV3, the backbone of DLLM, with minimal differences.
>
> We suggest the reviewer to refer to Section 4.3 of our paper. The encoder-decoder is used to generate goal states regards to the LLM instructions, which can hardly be similar from design to the encoder of DreamerV3.
>
>
> > CFG policy in Section 4.4.
>
> We suggest the reviewer to refer to Section 4.4 of our paper for the implementation and relevant hyperparameters of the CFG policy. In detail, we use guidance scale $\lambda = 4$ and use 'no achievement is reached' as the unconditional case.
>
> Since CFG is actually a biased sampling process when guidance strength $\lambda \neq 1$ for diffusion models and non-diffusion models, we only use it as a sampling method to show that the high-level policy actually learns to generate meaningful goals regards to the LLM instructions.
>
> > The results for baseline experiments in "LanGoal" match those in DLLM without proper citation.
>
> We apologize for missing the reference of baseline experiments.
> We have added the reference in the revised version, with results of LLM based methods and AdaRefiner from AdaRefiner, RL methods from Crafter benchmark, PPO (ResNet) and Dynalang from DLLM [3].
>
>
> We would like to clarify that, our work and DLLM differ in motivation and respective method on processing the LLM instructions. Our work and DLLM both consider exploration in sparse reward environments, but our work also aims to generate goal states regards to the LLM instructions. We thus query LLM to only generate 1 instruction each time, and add the exploration reward with LLM reward to encourage the high-level policy to generate goal states regards to the LLM instructions while exploring the environment.
>
> References:
>
> [1] Danijar Hafner et al. "Mastering diverse domains through world models." arXiv preprint arXiv:2301.04104, 2023.
>
> [2] Jessy Lin et al. "Learning to model the world with language." In Forty-first International Conference on Machine Learning, 2024.
>
> [3] Liu, Zeyuan, et al. "World Models with Hints of Large Language Models for Goal Achieving." arXiv preprint arXiv:2406.07381 (2024).

---

> ### Comment · Area_Chair_9MmJ · 2024-11-25
> **Please read rebuttal**
>
> Dear Reviewer hsev, Could you please read the authors' rebuttal and give them feedback at your earliest convenience? Thanks. AC

---

### Note · Authors · 2024-11-25

**Comment:**

We would like to withdraw our submission and refine our work to strengthen its contributions. We thank the reviewers for their constructive suggestions.

**Withdrawal Confirmation:**

I have read and agree with the venue's withdrawal policy on behalf of myself and my co-authors.